# Impact of early nutrition and feeding route on clinical outcomes of neurocritically ill patients

**Young Kyun Choi[1], Hyun-Jung Kim[2], Joonghyun Ahn[3], Jeong-Am Ryu[1,4]***

**1** Department of Critical Care Medicine, Samsung Medical Center, Sungkyunkwan University School of Medicine, Seoul, Republic of Korea, **2** Department of Dietetics, Samsung Medical Center, Seoul, Republic of Korea, **3** Statistic and Data Center, Clinical Research Institute, Samsung Medical Center, Seoul, Republic of Korea, **4** Department of Neurosurgery, Samsung Medical Center, Sungkyunkwan University School of Medicine, Seoul, Republic of Korea

* lamyud.ryu@samsung.com

**Data Availability Statement:** Data Sharing Statement: Our data are available on Harvard Dataverse Network (http://dx.doi.org/10.7910/DVN/3F8WF1).

## Abstract

Early proper nutritional support is important to critically ill patients. Nutritional support is also associated with clinical outcomes of neurocritically ill patients. We investigate whether early nutrition is associated with clinical outcomes in neurocritically ill patients. This was a retrospective, single-center, observational study including neurosurgical patients who were admitted to the intensive care unit (ICU) from January 2013 to December 2019. Patients who started enteral nutrition or parenteral nutrition within 72 hours after ICU admission were defined as the early nutrition group. The primary endpoint was in-hospital mortality. The secondary endpoint was an infectious complication. Propensity score matching (PSM) and propensity score weighting overlap weights (PSOW) were used to control selection bias and confounding factors. Among 1,353 patients, early nutrition was performed in 384 (28.4%) patients: 152 (11.2%) early enteral nutrition (EEN) and 232 (17.1%) early parenteral nutrition (EPN). In the overall study population, the rate of in-hospital mortality was higher in patients with late nutrition than in those with early nutrition (P<0.001). However, there was no significant difference in in-hospital mortality and infectious complications incidence between the late and the early nutrition groups in the PSM and PSOW adjusted population (all P>0.05). In the overall study population, EEN patients had a low rate of in-hospital mortality and infectious complications compared with those with EPN and late nutrition (P<0.001 and P = 0.001, respectively). In the multivariable analysis of the overall, PSM adjusted, and PSOW adjusted population, there was no significant association between early nutrition and in-hospital mortality and infectious complications (all P>0.05), but EEN was significantly associated with in-hospital mortality and infectious complications (all P<0.05). Eventually, early enteral nutrition may reduce the risk of in-hospital mortality and infectious complications in neurocritically ill patients.

## Introduction

Nutrition support plays an important role in the management of critically ill patients [1–3]. Malnutrition is associated with poor clinical outcomes such as higher rates of mortality (32%

**Funding:** Unfunded studies The authors received no specific funding for this work.

**Competing interests:** NO authors have competing interests The authors have declared that no competing interests exist.

vs. 14%, P = 0.018) [4], nosocomial infection (23.4% vs. 3.5%, P<0.001) [5], and long stay in intensive care unit (ICU) [4, 6]. Similarly, in critically ill patients with stroke or traumatic brain injury, nutritional support is associated with neurological prognosis and mortality [7–9]. In stroke patients, the mortality rate of malnourished patients was 37%, which was significantly higher than that of patients with normal nutrition which was 21% (P<0.001) [9]. In traumatic brain injury, it was reported that the rate of infection was reduced in early nutrition compared to delayed nutrition (risk ratio: 0.77, P = 0.04) [8]. Patients with brain injury commonly suffer from hypermetabolic reactions that can lead to increased energy and protein expenditure. Therefore, early nutrition may help to improve neurological prognosis [10, 11].

However, it is not easy to focus on nutrition in the early stage of neurocritical illness. Moreover, nutrition support is often underestimated and considered a lower priority than maintaining cerebral perfusion pressure and other medical problems in neurocritical ill patients [8, 12]. In addition, the optimal feeding timing, route, and formula in these patients are still unclear [8, 13]. Therefore, the objective of this study was to investigate whether early nutrition was associated with clinical outcomes in patients who were admitted to the neurosurgical intensive care unit (ICU) and to determine the optimal feeding timing, route, and formula. The early nutrition group was split into early enteral nutrition (EEN) and early parenteral nutrition (EPN) groups and the effects of EEN were investigated. In addition, we evaluated whether early nutrition *per se* was associated with poor prognosis when severity and factors other than nutritional support were controlled by propensity score matching (PSM) and propensity score weighting overlap weights (PSOW).

## Materials and methods

### Study population

This was a retrospective, single-center, observational study. Patients who were admitted to the neurosurgical ICU at the Samsung Medical Center, Seoul, Republic of Korea, tertiary referral hospital from January 2013 to December 2019 were eligible. This study was approved by the Institutional Review Board (IRB) of the Samsung Medical Center (IRB approval number: SMC 2020-09-082). Patients' records were reviewed and published according to the Declaration of Helsinki. The requirement of informed consent was waived by the IRB due to its retrospective nature. We included patients who were hospitalized in the neurosurgical ICU for more than 5 days due to neurocritical illness or neurosurgical postoperative management. We excluded patients who had insufficient medical records, who had a 'do not resuscitation' order, who were admitted to departments other than neurosurgery, and who were transferred to other hospitals or with unknown prognoses (Fig 1).

### Definitions and endpoints

In this study, baseline characteristics such as comorbidities, behavioral risk factors, ICU management, and laboratory data were collected retrospectively using a Clinical Data Warehouse. Our center constructed a "Clinical Data Warehouse Darwin-C" designed for searching and retrieving de-identified medical records from electronic archives. It contains data for more than four million patients.

Patients who started enteral nutrition or parenteral nutrition within 72 hours after ICU admission were defined as the early nutrition group [8, 10] which was further divided into EEN and EPN groups. Infectious complications were defined as nosocomial infections such as pneumonia, central nervous system infection, bloodstream infection, urinary tract infection, and sepsis [8, 10]. The primary endpoint was in-hospital mortality while the secondary endpoint was an infectious complication.

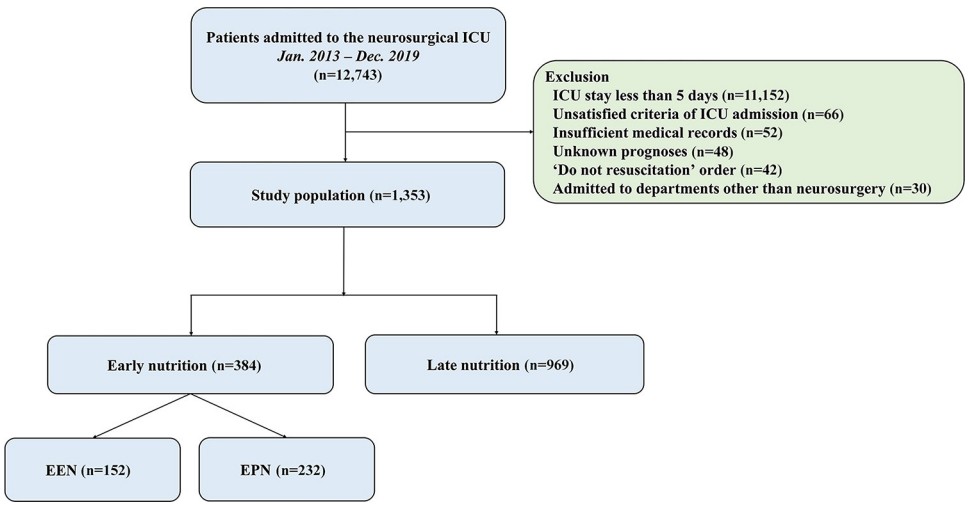

**Fig 1. Study flow chart.** ICU, intensive care unit; EEN, early enteral nutrition; EPN, early parenteral nutrition.

## Statistical analyses

All data are presented as means ± standard deviations for continuous variables or frequencies and proportions for categorical variables. Data were compared using Student's *t*-test and one-way analysis of variance for continuous variables and Chi-square test or Fisher's exact test for categorical variables. In this study, patients with EEN were relatively few compared to those with late nutrition. In addition, the severity scores of patients differed for each nutritional group. Therefore, we used several analysis methods to control various biases arising from these differences. PSM and PSOW were used to control for selection bias and confounding factors [14]. In PSM analysis, each patient with early nutrition or EEN was matched to one control patient with the nearest neighbor matching within calipers determined by the propensity score. A caliper width of 0.2 of the standard deviation of the logit of the propensity score was used for the matching [15]. We compared the balance of baseline covariates between nutrition groups by calculating the standardized mean difference (SMD) [16]. If PSM and PSOW methods were effective for balancing exposure groups, the SMD should be close to zero [17]. Therefore, SMDs of less than 10% were used for proper balancing between the two groups. To evaluate whether there were differences in in-hospital mortality and infectious complications according to nutrition patterns, we performed multiple logistic regression with stepwise variable selection in the overall, PSM, and PSOW population. In the overall population, we tried to obtain results after correcting confounding through regression adjustment. In addition, we performed a doubly robust estimation to additionally correct the bias that might still exist after PSM and PSOW. Variables included in the multiple analyses were age, sex, comorbidities, cause of ICU admission, utilization of organ support modalities (including mechanical ventilators, continuous renal replacement therapy, and vasopressors, intracranial pressure (ICP) monitoring devices, and hyperosmolar therapy), Glasgow Coma Scale (GCS) and Acute Physiology and Chronic Health Evaluation (APACHE) II score on ICU admission, and/or early nutrition, EEN, and EPN. Since there might be biases arising from substantial subject loss after PSM and biased weight due to the misspecified PSOW model in this study, it was necessary to verify the robustness of the results of all the analysis methods. All the tests were two-sided and *p* values of less than 0.05 were considered statistically significant. All statistical analyses were performed with R Statistical Software version 4.2.0 (R Foundation for Statistical Computing, Vienna, Austria).

## Results

### Baseline characteristics

A total of 12,743 patients were admitted to the neurosurgical ICU during the study period and 1,353 patients were included in the final analysis. In the overall study population, early nutrition was performed in 384 (28.4%) patients (Fig 1); EEN in 152 (11.2%) patients and EPN in 232 (17.1%) patients. The mean age of all the patients was 50.5 ± 23.2 years. There were 707 (52.3%) male patients. Malignancy (55.3%) and hypertension (34.5%) were the most common comorbidities. Brain tumors (37.5%) and intracerebral hemorrhage (17.4%) were the most common reasons for ICU admission (Table 1).

### Clinical outcomes

**In-hospital mortality.** In the overall study population, the rate of in-hospital mortality was higher in patients with late nutrition than in those with early nutrition (33.1% vs. 14.1%, $P < 0.001$) (Table 1). Rates of in-hospital mortality were also different between EEN, EPN, and late nutritional groups ($P < 0.001$) (Table 2). However, such difference might be due to differences in age, causes of ICU admission, and severity scores (all $P < 0.001$) among these three nutritional groups (Table 2).

SMDs of covariates were used for propensity score modeling before and after PSM and PSOW adjustments. After adjustment, most covariates showed SMDs within the 10% cutoff (Fig 2). Unlike the overall study population, there were no significant differences in the rates of in-hospital mortality between the late nutrition group and the early nutrition group of the PSM and PSOW adjusted population ($P = 0.234$ and $P = 0.094$, respectively) (S1 Table). In multivariable analyses of the overall, PSM adjusted, and PSOW adjusted population, early nutrition was not significantly associated with in-hospital mortality (all $P > 0.05$), but EEN was significantly associated with in-hospital mortality (all $P < 0.05$) (Fig 3A) (S2 Table).

### Infectious complications

There was no significant difference in infectious complications between the late nutrition group and the early nutrition group in the overall study population and the PSM and PSOW adjusted population (all $P > 0.05$) (Table 1 and S1 Table). However, rates of infectious complications were different between EEN, EPN, and the late nutritional groups ($P = 0.001$) (Table 2 and S2 Table). In multivariable analyses of the overall, PSM adjusted, and PSOW adjusted population, early nutrition was not significantly associated with infectious complications (all $P > 0.05$), but EEN was significantly associated with infectious complications (all $P < 0.05$) (Fig 3B).

## Discussion

In this study, we investigated whether early nutrition was associated with clinical outcomes in patients admitted to neurosurgical ICU. The major findings of this study were as follows. First, early nutrition was performed in approximately one-third of neurocritically ill patients, and two-fifth of early nutrition was administered as EEN. Second, early nutrition, including EEN showed an association with clinical outcomes of neurocritically ill patients in univariable analysis. However, the numbers of patients with EEN and EPN were small and severity levels were different between the nutrition groups. Finally, in the overall, PSM adjusted, and PSOW adjusted population, multivariable analyses revealed that early nutrition was not significantly associated with in-hospital mortality and infectious complications, but EEN was significantly associated with in-hospital mortality and infectious complications.

**Table 1. Baseline characteristics according to the timing of nutrition.**

| | Overall study population | | | |
|---|---|---|---|---|
| | Late nutrition (n = 969) | Early nutrition (n = 384) | P value | SMD |
| Patient demographics | | | | |
| Age (year) | 50.1 ± 23.7 | 51.66 ± 22.1 | 0.267 | 0.068 |
| Sex, male | 512 (52.8) | 195 (50.8) | 0.534 | 0.041 |
| Comorbidities | | | | |
| Malignancy | 528 (54.5) | 220 (57.3) | 0.382 | 0.056 |
| Hypertension | 332 (34.3) | 135 (35.2) | 0.804 | 0.019 |
| Diabetes mellitus | 131 (13.5) | 58 (15.1) | 0.502 | 0.045 |
| Chronic kidney disease | 64 (6.6) | 32 (8.3) | 0.318 | 0.066 |
| Cardiovascular disease | 42 (4.3) | 10 (2.6) | 0.182 | 0.095 |
| Chronic liver disease | 30 (3.1) | 16 (4.2) | 0.416 | 0.057 |
| Behavioral risk factors | | | | |
| Current alcohol consumption | 199 (20.5) | 86 (22.4) | 0.495 | 0.045 |
| Current smoking | 98 (10.1) | 52 (13.5) | 0.086 | 0.106 |
| Cause of ICU admission | | | 0.002 | 0.295 |
| Brain tumor | 351 (36.2) | 157 (40.9) | | |
| Intracerebral hemorrhage | 179 (18.5) | 56 (14.6) | | |
| Traumatic brain injury | 152 (15.7) | 39 (10.2) | | |
| Subarachnoid hemorrhage | 122 (12.6) | 50 (13.0) | | |
| Elective vascular surgery | 72 (7.4) | 37 (9.6) | | |
| Cerebral infarction | 22 (2.3) | 11 (2.9) | | |
| Spinal surgery | 17 (1.8) | 12 (3.1) | | |
| Central nervous system infection | 12 (1.2) | 13 (3.4) | | |
| Others | 42 (4.3) | 9 (2.3) | | |
| APACHE II score on ICU admission | 8.3 ± 7.7 | 6.13 ± 5.5 | <0.001 | 0.326 |
| Glasgow coma scale on ICU admission | 11.8 ± 4.4 | 13.7 ± 2.5 | <0.001 | 0.549 |
| ICU management | | | | |
| Mechanical ventilation | 652 (67.3) | 207 (53.9) | <0.001 | 0.276 |
| Continuous renal replacement therapy | 39 (4.0) | 10 (2.6) | 0.272 | 0.079 |
| ICP monitoring | 407 (42.0) | 182 (47.4) | 0.081 | 0.109 |
| Use of mannitol* | 406 (41.9) | 167 (43.5) | 0.636 | 0.032 |
| Use of glycerin* | 391 (40.4) | 151 (39.3) | 0.775 | 0.021 |
| Use of vasopressors | 160 (16.5) | 45 (11.7) | 0.033 | 0.138 |
| Clinical outcomes† | | | | |
| In-hospital mortality | 321 (33.1) | 54 (14.1) | <0.001 | |
| 28-day mortality | 295 (30.4) | 46 (12.0) | <0.001 | |
| ICU mortality | 281 (29.0) | 38 (9.9) | <0.001 | |
| ICU length of stay (hour) | 292.1 ± 769.3 | 329.7 ± 989.9 | 0.457 | |
| Hospital length of stay (day) | 68.9 ± 253.3 | 78.2 ± 177.9 | 0.511 | |
| Infectious complications | 82 (8.5) | 32 (8.3) | 0.999 | |

Data are presented as numbers (%) or means ± standard deviations.

*Some patients received more than one hyperosmolar agent.

†Variables are not retained in the propensity score matching

SMD, standardized mean difference; APACHE II, Acute Physiology and Chronic Health Evaluation; ICP, intracranial pressure, ICU, intensive care unit; ICP, intracranial pressure.

**Table 2. Baseline characteristics of patients with late nutrition, early enteral and early parenteral feeding.**

| | Overall study population | | | | |
|---|---|---|---|---|---|
| | Non-EEN (n = 1201) | | EEN (n = 152) | P value | SMD |
| | Late nutrition (n = 969) | EPN (n = 232) | | | |
| Patient demographics | | | | | |
| Age (year) | 50.1 ± 23.7 | 56.7 ± 17.6 | 44.0 ± 25.7 | <0.001 | 0.379 |
| Sex, male | 512 (52.8) | 120 (51.7) | 75 (49.3) | 0.714 | 0.047 |
| Comorbidities | | | | | |
| Malignancy | 528 (54.5) | 122 (52.6) | 98 (64.5) | 0.047 | 0.162 |
| Hypertension | 332 (34.3) | 90 (38.8) | 45 (29.6) | 0.171 | 0.130 |
| Diabetes mellitus | 131 (13.5) | 44 (19.0) | 14 (9.2) | 0.020 | 0.189 |
| Chronic kidney disease | 64 (6.6) | 19 (8.2) | 13 (8.6) | 0.531 | 0.049 |
| Cardiovascular disease | 42 (4.3) | 6 (2.6) | 4 (2.6) | 0.328 | 0.064 |
| Chronic liver disease | 30 (3.1) | 8 (3.4) | 8 (5.3) | 0.390 | 0.072 |
| Behavioral risk factors | | | | | |
| Current alcohol consumption | 199 (20.5) | 63 (27.2) | 23 (15.1) | 0.014 | 0.198 |
| Current smoking | 98 (10.1) | 39 (16.8) | 13 (8.6) | 0.008 | 0.167 |
| Cause of ICU admission | | | | <0.001 | 0.490 |
| Brain tumor | 351 (36.2) | 85 (36.6) | 72 (47.4) | | |
| Intracerebral hemorrhage | 179 (18.5) | 42 (18.1) | 14 (9.2) | | |
| Traumatic brain injury | 152 (15.7) | 34 (14.7) | 5 (3.3) | | |
| Subarachnoid hemorrhage | 122 (12.6) | 28 (12.1) | 22 (14.5) | | |
| Elective vascular surgery | 72 (7.4) | 14 (6.0) | 23 (15.1) | | |
| Cerebral infarction | 22 (2.3) | 9 (3.9) | 2 (1.3) | | |
| Spinal surgery | 17 (1.8) | 8 (3.4) | 4 (2.6) | | |
| Central nervous system infection | 12 (1.2) | 6 (2.6) | 7 (4.6) | | |
| Others | 42 (4.3) | 6 (2.6) | 3 (2.0) | | |
| APACHE II score on ICU admission | 8.3 ± 7.7 | 6.4 ± 6.1 | 5.8 ± 4.4 | <0.001 | 0.264 |
| Glasgow coma scale on ICU admission | 11.8 ± 4.4 | 13.2 ± 2.9 | 14.5 ± 1.3 | <0.001 | 0.604 |
| ICU management | | | | | |
| Mechanical ventilation | 652 (67.3) | 155 (66.8) | 52 (34.2) | <0.001 | 0.467 |
| Continuous renal replacement therapy | 39 (4.0) | 9 (3.9) | 1 (0.7) | 0.115 | 0.150 |
| ICP monitoring | 407 (42.0) | 105 (45.3) | 77 (50.7) | 0.114 | 0.116 |
| Use of mannitol* | 406 (41.9) | 93 (40.1) | 74 (48.7) | 0.216 | 0.116 |
| Use of glycerin* | 391 (40.4) | 119 (51.3) | 32 (21.1) | <0.001 | 0.437 |
| Use of vasopressors | 160 (16.5) | 29 (12.5) | 16 (10.5) | 0.075 | 0.117 |
| Clinical outcomes[†] | | | | | |
| In-hospital mortality | 321 (33.1) | 45 (19.4) | 9 (5.9) | <0.001 | |
| 28-day mortality | 295 (30.4) | 38 (16.4) | 8 (5.3) | <0.001 | |
| ICU mortality | 281 (29.0) | 33 (14.2) | 5 (3.3) | <0.001 | |
| ICU length of stay (hour) | 292.1 ± 769.3 | 298.1 ± 265.6 | 377.8 ± 1540.6 | 0.501 | |
| Hospital length of stay (day) | 68.9 ± 253.3 | 67.0 ± 81.9 | 95.2 ± 263.7 | 0.415 | |
| Infectious complications | 82 (8.5) | 29 (12.5) | 3 (2.0) | 0.001 | |

Data are presented as numbers (%) or means ± standard deviations.

Data show a comparison between late nutrition, EPN, and EEN.

*Some patients received more than one hyperosmolar agent.

[†]Variables are not retained in the propensity score matching.

EEN, early enteral nutrition; EPN, early parenteral nutrition; SMD, standardized mean difference; APACHE II, Acute Physiology and Chronic Health Evaluation; ICU, intensive care unit; ICP, intracranial pressure.

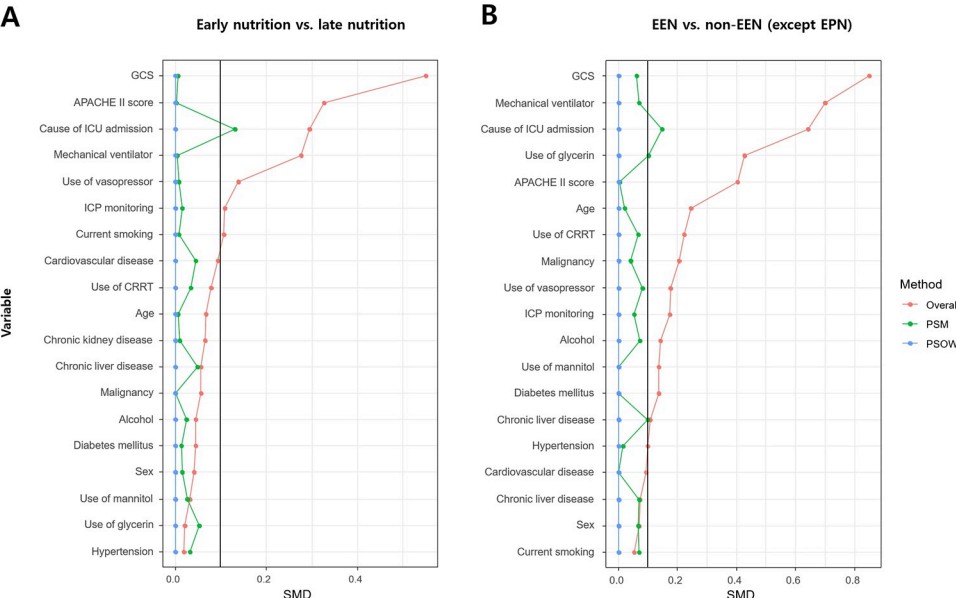

**Fig 2.** Standardized mean differences (SMDs) between nutrition groups (early nutrition vs. late nutrition [A] and early enteral nutrition [EEN] vs. non-EEN [B]) according to propensity score matching (PSM) and propensity score weighting overlap weights (PSOW). The balance of baseline covariates between nutrition groups were compared by calculating the SMD. If PSM and PSOW methods were effective for balancing the exposure groups, the SMD should be less than 10% as proper balancing between the two groups. After adjustment, most covariates showed SMDs within the 10% cutoff. GCS, Glasgow Coma Scale; APACHE II, Acute Physiology and Chronic Health Evaluation; ICU, intensive care unit; ICP, intracranial pressure; CRRT, continuous renal replacement therapy.

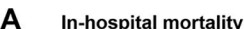

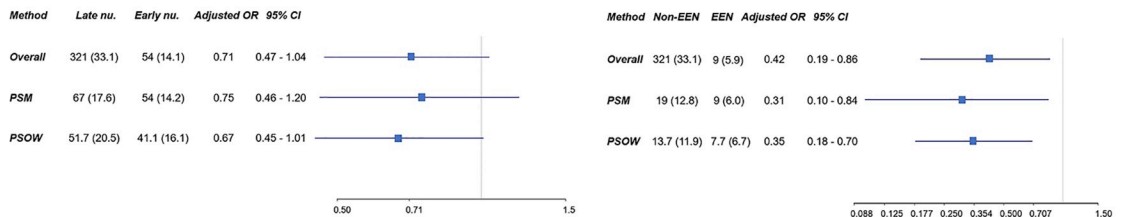

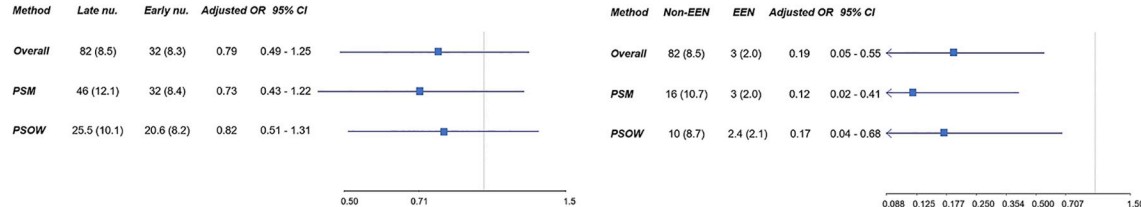

**Fig 3.** In multivariable analyses of the overall population, propensity score matching (PSM) adjusted population, and propensity score weighting overlap weights (PSOW) population, early nutrition was not significantly associated with in-hospital mortality, but EEN was significantly associated with in-hospital mortality (A). In addition, early nutrition was not significantly associated with infectious complications, but EEN was significantly associated with infectious complications (B). Non-EEN means only late nutrition without EPN. nu., nutrition; OR, odds ratio; CI, confidence interval; EEN, early enteral nutrition; EPN, early parenteral nutrition.

In the early stages of neurocritically ill patients, appropriate nutritional support is important due to hypermetabolic responses after brain injury [10, 11]. However, sympathetic hyperactivation arising from increased intracranial pressure can affect gastrointestinal function [18–20]. Moreover, early intragastric feeding can increase the risk of gastric residual volume, delayed gastric emptying, and aspiration pneumonia in neurocritically ill patients [12]. Although there has been a lot of debate about the optimal timing and the route of feeding [8, 13], a recent meta-analysis has shown that EPN is superior to EEN in reducing mortality and infectious complications and improving outcomes of patients with traumatic brain injury in the acute gut-intolerant phase [8]. However, in the present study, early nutrition did not affect clinical prognosis. Moreover, EEN, rather than EPN, was associated with decreased mortality and nosocomial infections. EPN may be associated with delayed recovery and more complications, as compared with late parenteral nutrition [21]. Therefore, combining the EEN group with the EPN group, or the late nutrition group with the EPN group might not be useful in evaluating the association between feeding options and clinical outcomes. EPN might affect the outcome as a confounding factor in this study.

EEN has several benefits in the treatment of critically ill patients [22–26]. First, gastrointestinal tract plays an important role in the immune responses [27]. However, the immune function of the gastrointestinal tract is disturbed in the early stage of critically ill patients [28]. In addition, pathogenic bacterial translocation of the gastrointestinal tract can stimulate systemic cytokine release and increase susceptibility to infections [27]. These changes can lead to multiple organ dysfunction and poor clinical outcomes [10, 27]. In the early stages of critically ill patients, enteral nutrition can maintain gastrointestinal integrity and prevent intestinal bacterial translocation [24, 29]. Second, EEN can enhance recovery in the early hypermetabolic stage of patients with multiple traumas including brain injury [26, 30, 31]. Third, enteral feeding is more physiologic, less invasive, and less expensive than total parenteral nutrition [32]. Therefore, EEN is associated with favorable outcomes in critically ill patients [32]. Recent studies have also shown that EEN can reduce rates of mortality and infectious complications in patients with intracranial hemorrhage and traumatic brain injury [26, 29, 30].

Nutritional support could be ignored in the early stages of patients with severely injured brains as the neurocritical or critical issues, including cerebral blood flow, hemodynamic instability, and lung injury, are more focused on than nutrition in these patients [8, 12]. Consequently, it is difficult to provide appropriate nutrition to critical patients in the early stage. Therefore, malnutrition can occur more easily in patients with severe neurological diseases than in those with benign diseases. It is not easy to determine whether late nutrition or inappropriate nutrition is associated with a poor prognosis since severe brain-injured patients generally have poor prognosis. Therefore, PSM and PSOW methods were used to adjust for this confounder in this study. Eventually, EEN was found to be significantly associated with favorable clinical outcomes in neurocritically ill patients.

Adequate calorie and protein intake is important for recovery in critically ill patients [33, 34]. Adequate nutritional support may also be important in neurocritically ill patients. The patients with severe traumatic brain injury have increased energy expenditure usually increase by 87%–200% above the usual requirement and may be elevated for 30 days due to metabolic changes [26, 35, 36]. In addition, systemic catabolic change could lead to hyperglycemia, protein wasting, and increased calorie demands [26, 35]. Therefore, optimized calorie and protein supply is also important for acute brain injury patients. However, energy expenditure, calorie and protein intake were not considered in this study. Accurate analysis of calorie and protein supply and energy expenditure may be necessary to investigate the relationship between early nutrition and clinical outcomes in neurocritically ill patients.

This study has several limitations. First, this was a retrospective review of medical records using data extracted from a Clinical Data Warehouse. The nonrandomized nature of registry data might have resulted in a selection bias. Second, the amount of EEN or EPN calorie intake for patients in the early stage was not considered in this study due to its retrospective nature. Third, nutritional support was performed occasionally through non-protocol methods for neurocritically ill patients. Finally, the distribution of neurosurgical diseases differed from that of the general neurosurgical ICU and the proportion of patients with brain tumors was particularly high.

## Conclusions

In this study, EEN may reduce the risk of in-hospital mortality and infectious complications in neurocritically ill patients. In addition, timely and proper nutritional support may be important to improve clinical outcomes in neurocritically ill patients.

## Supporting information

**S1 Table. Baseline characteristics according to timing of nutrition in the overall, PSM and PSOW adjusted population.**
(DOCX)

**S2 Table. Baseline characteristics of patients with and without early enteral feeding in the overall, PSM and PSOW adjusted population.**
(DOCX)

## Acknowledgments

We would like to thank the nursing director of the neurosurgical intensive care unit, Suk Kyung Choo, for providing excellent advice and fruitful discussions. We would also like to thank all the nurses of the neurosurgical intensive care unit at the Samsung Medical Center for their support in the completion of this study.

## Author Contributions

**Conceptualization:** Young Kyun Choi, Joonghyun Ahn, Jeong-Am Ryu.

**Data curation:** Young Kyun Choi, Hyun-Jung Kim.

**Formal analysis:** Young Kyun Choi, Joonghyun Ahn, Jeong-Am Ryu.

**Investigation:** Young Kyun Choi, Hyun-Jung Kim, Jeong-Am Ryu.

**Methodology:** Jeong-Am Ryu.

**Project administration:** Jeong-Am Ryu.

**Resources:** Hyun-Jung Kim.

**Supervision:** Jeong-Am Ryu.

**Visualization:** Joonghyun Ahn, Jeong-Am Ryu.

**Writing – original draft:** Young Kyun Choi, Joonghyun Ahn, Jeong-Am Ryu.

**Writing – review & editing:** Jeong-Am Ryu.

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
