## [Decision Letter · Decision Letter 0]

10 Jun 2022

PONE-D-21-39517Impact of early nutrition and feeding route on clinical outcomes of neurocrically ill patientsPLOS ONE

Dear Dr. Ryu,

Thank you for submitting your manuscript to PLOS ONE. After careful consideration, we feel that it has merit but does not fully meet PLOS ONE’s publication criteria as it currently stands. Therefore, we invite you to submit a revised version of the manuscript that addresses the points raised during the review process.

We look forward to receiving your revised manuscript.

Kind regards,

Rishabh Charan Choudhary

Academic Editor

PLOS ONE

Journal Requirements:

Additional Editor Comments (if provided):

Dear Author, we have received the review report from the reviewers. The manuscript needs major revision. Please made the corrections and submit it.

Thanks

Reviewers' comments:

Reviewer's Responses to Questions

**Comments to the Author**

1. Is the manuscript technically sound, and do the data support the conclusions?

Reviewer #1: Partly

Reviewer #2: Partly

2. Has the statistical analysis been performed appropriately and rigorously? 

Reviewer #1: Yes

Reviewer #2: Yes

3. Have the authors made all data underlying the findings in their manuscript fully available?

Reviewer #1: Yes

Reviewer #2: Yes

4. Is the manuscript presented in an intelligible fashion and written in standard English?

Reviewer #1: Yes

Reviewer #2: Yes

5. Review Comments to the Author

Reviewer #1: Comments:

The authors have provided a very thorough analysis of a retrospective dataset and have added to the body of literature on this important subject. They have also taken into consideration the limitations of this type of study.

Concerns:

In Table 2, it is not clear what the p-values indicate. The legend should indicate what comparison is being evaluated. Is it the comparison between the EEN and the Non-EEN patients or the comparison between the EEN and the EPN patients or between the early nutrition and the late nutrition patients? Based on the title of the table, I would presume that it is the comparison between the EEN and the Non-EEN patients.

In general, the sample size for this study is rather small for the number of variables considered in the analysis. As the authors noted, the number of patients with EEN and with EPN were small and the severity of illness is notably different between these two groups and the late nutrition group of patients.

Other investigators have found that mortality was higher in critically ill children who received parenteral nutrition or that late parenteral nutrition was associated with better outcomes than early parenteral nutrition in critically ill children. Thus, combining the EEN and the EPN groups might not be beneficial. In fact, including the EPN group with the late nutrition group might not be beneficial, particularly for such a small number of patients.

Other factors which should be considered would be the amount of nutrition provided to patients with respect to caloric intake and protein intake. The timing of nutrition may not be as important as the quantity and the composition of the nutrition provided. However, even the timing may impact outcomes. Why did the investigators choose to define early nutrition to be any nutrition before 72 hours? Did they consider 48 hours or even 24 hours?

Finally, I believe the investigators provided the various methods of data analysis to demonstrate a thorough approach, which is impressive, but presenting all analyses is probably unnecessary. Which analytic method is the best for this type of study? Which method retains the most data and most appropriately fits this study design and addresses the concerns of selection bias and confounding by comorbidities, clinical diagnoses, and severity of illness. This should have been determined in advance.

I would recommend revising the analysis to compare the quantity and the composition of the nutrition provided and I would consider excluding those who received EPN from the comparison of EEN to non-EEN. I would also limit the different types of analyses to reflect the most appropriate use of the data.

Reviewer #2: Dear authors,

I congratulate you for this paper. The statistical analysis is robust; however, the manuscript will benefit from major changes before publication.

The authors used retrospective data to evaluate the impact of early nutritional therapy and mortality. Also they analyzed the route of nutritional therapy.

First, I suggest an english review of the manuscript - it is understandable, but difficult to read.

An example in the background:

"Nutritional support is also associated with neurological prognosis and mortality of neurocritically ill patients with stroke or traumatic brain injury [6-8]. Early nutrition is especially important because it can affect neurological prognosis."

Associated with increased risk of mortality or a reduction in mortality? can affect neurological prognosis in what way - good or bad? And this same kind of sentence is repeated throughout the manuscript.

I will comment each part of the manuscript separately.

Abstract:

Background - there is no background information, only the objective of the study. Please include a sentence that explains the current knowledge.

Methods - no study design (retrospective cohort…)

Conclusion - can be clearer - again english editing.

Background:

Better review of literature. Bring the hard data from previous study to support your argument (meaning mortality reduction in % (p=...) or no statistical difference in other study (p =...)). Make it clear what new information this study adds

Methods:

The methods and statistical analysis are clear

Result:

The results are confusing.

I suggest dividing the results into the outcomes: primary (mortality) and secondary (infections complications) and not according statistical analysis.

Also, table 2 - the p is for the comparasion between the 3 groups (late nutrition, EPN and EEN) or between (non-EEN and EEN). It is not clear.

5.Discussion:

The comparison between current literature and the study result has room to be improved, but mostly English editing will help a lot.

There was a lot a of EPN. Any reason why? It is not supported by any guidelines. This should be better addressed in the discussion.

Do you have data regarding calories and protein intake in each group? I imagine that those in the early group would have a lesser caloric and protein deficit

6. PLOS authors have the option to publish the peer review history of their article (what does this mean?). If published, this will include your full peer review and any attached files.

Reviewer #1: No

Reviewer #2: **Yes: **Marina Verçoza Viana

---

## [Author Response · Author response to Decision Letter 0]

13 Jul 2022

July 11 2022

Dr. Rishabh Charan Choudhary 

PLOS ONE

Manuscript ID: PONE-D-21-39517

Title: Impact of early nutrition and feeding route on clinical outcomes of neurocritically ill patients

Dear Dr. Rishabh Charan Choudhary

Thank you very much for your letter and for the helpful comment from the reviewer. We appreciate the opportunity to resubmit our revised manuscript entitled “Impact of early nutrition and feeding route on clinical outcomes of neurocritically ill patients”. As always, you and your editorial staff have again provided us with a comprehensive and prompt review. Many of the valuable and constructive points that the reviewers pointed out were well taken by all the authors. After going over the reviewer’s comments, my colleagues and I have performed additional investigation and made some revisions in hopes of improving our paper. The revised and added portions of the manuscript are stated in the “Response to Reviewers” and are underlined and highlighted in the revised manuscript for your convenience.

All authors contributed to the conception and interpretation of data, drafting of the manuscript, revising it critically for important intellectual content, and final approval of the manuscript. The whole manuscript or part of it, neither has been published and is not being considered for publication elsewhere in any language except as an abstract. None of the authors have any financial relationships with any company or any other bias or conflict of interest.

We believe that these findings have scientific and clinical impact and will be interesting and informative to your readers. We hope that, upon review, our study will be found to be meritorious of publication in the PLOS ONE.

Yours sincerely,

Jeong-Am Ryu, M.D., Ph.D.

Department of Critical Care Medicine and Department of Neurosurgery, Samsung Medical Center, Sungkyunkwan University School of Medicine, 81 Irwon-ro, Gangnam-gu, Seoul 06351, Republic of Korea 

Tel: 82-2-3410-6399, Fax: 82-2-2148-7088

E-mail: lamyud.ryu@samsung.com

Response to Reviewers

Reviewer #1: 

Comments: The authors have provided a very thorough analysis of a retrospective dataset and have added to the body of literature on this important subject. They have also taken into consideration the limitations of this type of study.

Concerns:

In Table 2, it is not clear what the p-values indicate. The legend should indicate what comparison is being evaluated. Is it the comparison between the EEN and the Non-EEN patients or the comparison between the EEN and the EPN patients or between the early nutrition and the late nutrition patients? Based on the title of the table, I would presume that it is the comparison between the EEN and the Non-EEN patients.

R. We apologize for the lack of clear statement. We compared late nutrition, EPN and EEN. We wanted to show which of the late nutrition, EPN, and EEN was the better form of nutrition. As your recommendation, we added the following legend indicated what comparison to revised table 2. 

Data show a comparison between late nutrition, EPN and EEN.

In general, the sample size for this study is rather small for the number of variables considered in the analysis. As the authors noted, the number of patients with EEN and with EPN were small and the severity of illness is notably different between these two groups and the late nutrition group of patients.

R. We agree with the reviewer’s comment. There were differences in the number of subjects and the severity of illness among the nutrition groups. In this study, there were small sample size of subjects with EEN or EPN. In addition, it would be difficult to provide appropriate nutrition to neurocritically ill patients in the early stage. Nutritional support may be often ignored in early stages of patients with severely injured brain. Therefore, it is difficult to evaluate association between early proper nutrition and clinical outcomes of neurocritically ill patients. As a result, we performed various statistical methods to correct the bias. We evaluated whether early nutrition per se was associated with poor prognosis when severity and factors other than nutritional support were controlled by propensity score matching and propensity score weighting.

Other investigators have found that mortality was higher in critically ill children who received parenteral nutrition or that late parenteral nutrition was associated with better outcomes than early parenteral nutrition in critically ill children. Thus, combining the EEN and the EPN groups might not be beneficial. In fact, including the EPN group with the late nutrition group might not be beneficial, particularly for such a small number of patients.

R. We agree with the reviewer’s comment. Late parenteral nutrition may be associated with faster recovery and fewer complications, as compared with EPN (Ref. 21). Therefore, combining the EEN and the EPN groups, or the late nutrition and the EPN group might not be beneficial. EPN might affect the outcome as a confounding factor. As your recommendation, we re-analyzed the difference between EEN and non-EEN (only late nutrition, except EPN). We revised Figure 2 & 3 and added the following sentences in the Discussion section of the revised manuscript (line 214-218 in page 15). 

EPN may be associated with delayed recovery and more complications, as compared with late parenteral nutrition (Ref. 21). Therefore, combining the EEN group with the EPN group, or the late nutrition group with the EPN group might not be useful in evaluating the association between feeding options and clinical outcomes. EPN might affect the outcome as a confounding factor in this study.

Ref. 21: Casaer MP, Mesotten D, Hermans G, Wouters PJ, Schetz M, Meyfroidt G, et al. Early versus late parenteral nutrition in critically ill adults. N Engl J Med. 2011; 365: 506-17. https://doi.org/10.1056/NEJMoa1102662 PMID: 21714640

Other factors which should be considered would be the amount of nutrition provided to patients with respect to caloric intake and protein intake. The timing of nutrition may not be as important as the quantity and the composition of the nutrition provided. However, even the timing may impact outcomes. Why did the investigators choose to define early nutrition to be any nutrition before 72 hours? Did they consider 48 hours or even 24 hours?

R. We agree with the reviewer’s comment. In Dr. Wang’s study (Ref. 8), which was a well-designed meta-analysis, early nutrition was defined as 72 hours. In our study, early nutrition was also defined as feeding within 72 hours referring to the Dr. Wang’s study. Even the early nutrition, which was defined as feeding within 72 hours, had a smaller sample size in our study. Therefore, it was difficult to perform a statistical analysis with the definition of feeding within 24 or 48 hours because the sample sizes would be so small in these definitions. As a result, in this study, early nutrition was defined as feeding within 72 hours referring to the Dr. Wang’s study.

Ref. 8: Wang X, Dong Y, Han X, Qi XQ, Huang CG, Hou LJ. Nutritional support for patients sustaining traumatic brain injury: a systematic review and meta-analysis of prospective studies. PLoS One. 2013; 8: e58838. https://doi.org/10.1371/journal.pone.0058838 PMID: 23527035

In addition, there was the limitation of this study. Unfortunately, the amount of EEN or EPN calorie intake for patients in the early stage was not considered in this study due to its retrospective nature. We have mentioned these limitations in the limitation section of the revised manuscript.

The amount of EEN or EPN calorie intake for patients in the early stage was not considered in this study due to its retrospective nature.

Finally, I believe the investigators provided the various methods of data analysis to demonstrate a thorough approach, which is impressive, but presenting all analyses is probably unnecessary. Which analytic method is the best for this type of study? Which method retains the most data and most appropriately fits this study design and addresses the concerns of selection bias and confounding by comorbidities, clinical diagnoses, and severity of illness. This should have been determined in advance.

R. We agree with the reviewer’s comment. We performed various statistical methods to correct the bias. However, as your concerns, various statistical methods can confuse the readers. PSTW (a relatively rare method) and IPTW (an inappropriate balance, frequently over 0.1 of SMD) has been removed from the original manuscript. As following graph, severity of the subjects and factors other than nutritional support were well controlled by PSM and PSOW in this study. We revised Figure 2 & 3.

I would recommend revising the analysis to compare the quantity and the composition of the nutrition provided and I would consider excluding those who received EPN from the comparison of EEN to non-EEN. I would also limit the different types of analyses to reflect the most appropriate use of the data.

R. We agree with the reviewer’s comment. As reviewer’s recommendation, those who received EPN have been excluded from the comparison of EEN to non-EEN. We re-analyzed the difference between EEN and non-EEN (only late nutrition, except EPN). We revised Table 2. In addition, PSTW and IPTW has been removed from the original manuscript because of a relatively rare method or an inappropriate balance.

We thank the reviewer for valuable comments. Addressing them fully has significantly strengthened the manuscript. 

Reviewer #2: 

Dear authors,

I congratulate you for this paper. The statistical analysis is robust; however, the manuscript will benefit from major changes before publication.

The authors used retrospective data to evaluate the impact of early nutritional therapy and mortality. Also they analyzed the route of nutritional therapy.

First, I suggest an english review of the manuscript - it is understandable, but difficult to read.

An example in the background:

"Nutritional support is also associated with neurological prognosis and mortality of neurocritically ill patients with stroke or traumatic brain injury [6-8]. Early nutrition is especially important because it can affect neurological prognosis."

Associated with increased risk of mortality or a reduction in mortality? can affect neurological prognosis in what way - good or bad? And this same kind of sentence is repeated throughout the manuscript.

I will comment each part of the manuscript separately.

R. We appreciate valuable comments. We have revised the redundant phrase. In our study, neurological outcomes were not investigated. Primary endpoint was in-hospital mortality. A more accurate expression is “EEN may reduce the risk of mortality in neurocritically ill patients”. We changed conclusion as “EEN may reduce the risk of in-hospital mortality and infectious complications in neurocritically ill patients.” In addition, as your recommendation, we revised each part of the manuscript separately. 

Abstract:

Background - there is no background information, only the objective of the study. Please include a sentence that explains the current knowledge.

Methods - no study design (retrospective cohort…)

Conclusion - can be clearer - again english editing.

R. As your comments, we revised the manuscript. Revised sentences have been underlined and highlighted in the revised manuscript for your convenience. 

Background:

Better review of literature. Bring the hard data from previous study to support your argument (meaning mortality reduction in % (p=...) or no statistical difference in other study (p =...)). Make it clear what new information this study adds

R. As your comment, we revised the section of Background. We provided accurate data and p-value in the text as the following sentences (line 35-42 in page 5). 

Malnutrition is associated with poor clinical outcomes such as higher rates of mortality (32% vs. 14%, P=0.018) [Ref. 4], nosocomial infection (23.4% vs. 3.5%, P<0.001) [Ref. 5], and long stay in intensive care unit (ICU) [Ref. 4,6]. Similarly, in critically ill patients with stroke or traumatic brain injury, nutritional support is associated with neurological prognosis and mortality [Ref. 7-9]. In stroke patients, the mortality rate of malnourished patients was 37%, which was significantly higher than that of patients with normal nutrition which was 21% (P<0.001) [Ref. 9]. In traumatic brain injury, it was reported that the rate of infection was reduced in early nutrition compared to delayed nutrition (risk ratio: 0.77, P=0.04) [Ref. 8].

Methods:

The methods and statistical analysis are clear

R. We appreciated your kind comment. We performed various statistical methods to correct the bias. However, as other reviewer’s opinion, various statistical methods can confuse the readers. Therefore, PSTW (a relatively rare method) and IPTW (an inappropriate balance, frequently over 0.1 of SMD) has been removed from the original manuscript.

Result:

The results are confusing.

I suggest dividing the results into the outcomes: primary (mortality) and secondary (infections complications) and not according statistical analysis.

R. As your comment, we revised the section of Results (line 138-171). 

Also, table 2 - the p is for the comparasion between the 3 groups (late nutrition, EPN and EEN) or between (non-EEN and EEN). It is not clear.

R. We apologize for the lack of clear statement. We compared late nutrition, EPN, and EEN. We added the following legend indicated what comparison to revised table 2. 

Data show a comparison between late nutrition, EPN and EEN. (the legend of revised Table 2)

As other reviewer’s recommendation, we re-analyzed the difference between EEN and non-EEN (only late nutrition, except EPN). Late parenteral nutrition may be associated with faster recovery and fewer complications, as compared with EPN (Ref. 21). Therefore, combining the EEN and the EPN groups, or the late nutrition and the EPN group might not be useful to evaluate the association between nutrition types and clinical outcomes. EPN might affect the outcome as a confounding factor. We revised Figure 2 & 3.

Ref. 21: Casaer MP, Mesotten D, Hermans G, Wouters PJ, Schetz M, Meyfroidt G, et al. Early versus late parenteral nutrition in critically ill adults. N Engl J Med. 2011; 365: 506-17. https://doi.org/10.1056/NEJMoa1102662 PMID: 21714640

5.Discussion:

The comparison between current literature and the study result has room to be improved, but mostly English editing will help a lot.

There was a lot a of EPN. Any reason why? It is not supported by any guidelines. This should be better addressed in the discussion.

Do you have data regarding calories and protein intake in each group? I imagine that those in the early group would have a lesser caloric and protein deficit

R. We agree with the reviewer's comments. In the early stages, the sympathetic tone may be elevated in patients with elevated intracranial pressure (Ref. 8). Therefore, paralytic ileus, abdominal distension, and vomiting may not be uncommon in these patients. Hence, PN would be preferred occasionally compared with EN in our center.

There was the limitation of this study. Unfortunately, the amount of EEN or EPN calorie intake for patients in the early stage was not considered in this study due to its retrospective nature. We have mentioned these limitations in the limitation section of the revised manuscript.

The amount of EEN or EPN calorie intake for patients in the early stage was not considered in this study due to its retrospective nature.

Ref. 8: Wang X, Dong Y, Han X, Qi XQ, Huang CG, Hou LJ. Nutritional support for patients sustaining traumatic brain injury: a systematic review and meta-analysis of prospective studies. PLoS One. 2013; 8: e58838. https://doi.org/10.1371/journal.pone.0058838 PMID: 23527035

As your recommendation, our manuscript has been proofread by a professional English editing service. We thank the reviewer for valuable comments. Addressing them fully has significantly strengthened the manuscript.

---

## [Decision Letter · Decision Letter 1]

1 Sep 2022

PONE-D-21-39517R1Impact of early nutrition and feeding route on clinical outcomes of neurocritically ill patientsPLOS ONE

Dear Dr. Ryu,

Thank you for submitting your manuscript to PLOS ONE. After careful consideration, we feel that it has merit but does not fully meet PLOS ONE’s publication criteria as it currently stands. Therefore, we invite you to submit a revised version of the manuscript that addresses the points raised during the review process.

ACADEMIC EDITOR: Please insert comments here and delete this placeholder text when finished. Be sure to:Indicate which changes you require for acceptance versus which changes you recommendAddress any conflicts between the reviews so that it's clear which advice the authors should followProvide specific feedback from your evaluation of the manuscriptPlease ensure that your decision is justified on PLOS ONE’s publication criteria and not, for example, on novelty or perceived impact.

We look forward to receiving your revised manuscript.

Kind regards,

Rishabh Charan Choudhary

Academic Editor

PLOS ONE

Journal Requirements:

Additional Editor Comments (if provided):

Reviewers' comments:

Reviewer's Responses to Questions

**Comments to the Author**

1. If the authors have adequately addressed your comments raised in a previous round of review and you feel that this manuscript is now acceptable for publication, you may indicate that here to bypass the “Comments to the Author” section, enter your conflict of interest statement in the “Confidential to Editor” section, and submit your "Accept" recommendation.

Reviewer #1: (No Response)

Reviewer #2: All comments have been addressed

2. Is the manuscript technically sound, and do the data support the conclusions?

Reviewer #1: Yes

Reviewer #2: Yes

3. Has the statistical analysis been performed appropriately and rigorously? 

Reviewer #1: Yes

Reviewer #2: Yes

4. Have the authors made all data underlying the findings in their manuscript fully available?

Reviewer #1: Yes

Reviewer #2: Yes

5. Is the manuscript presented in an intelligible fashion and written in standard English?

Reviewer #1: Yes

Reviewer #2: Yes

6. Review Comments to the Author

Reviewer #1: Second Review

Thank you for the revised manuscript. It is much improved.

Concerns:

The legend for Table 2 clarifies the comparison. However, since there are three groups being compared it would seem that comparison of continuous variables would have required ANOVA rather than Student’s t-test. If this was done, please add this to the description of the data analysis. If this was not done, please explain what method of analysis was used to compare data for continuous variables in three groups.

In Figure 2, there are lines for the IPTW analysis, but you reported removing this analysis from the manuscript. It should be removed from the Figure as well.

Finally, I previously recommended that you compare the quantity and the composition of the nutrition provided but you indicated that these data are not available. You also indicated that you mentioned this as a limitation in your discussion section. I would encourage you to expand upon this in the discussion. Because the data for the quantity and the composition of the nutrition provided were not available, you are only comparing the timing of initiating nutrition by either the enteral or the parenteral route. However, the quantity of nutrition may have an impact on the findings of your study. Some previous studies have shown a difference in outcomes based on calories delivered, based on protein delivered, and some on both calories and protein delivered. This is an unfortunate deficit in your study.

Reviewer #2: Dear authors,

Thank you for your answers.

I do believe the manuscript is improved and all questions have been adequately answered.

7. PLOS authors have the option to publish the peer review history of their article (what does this mean?). If published, this will include your full peer review and any attached files.

Reviewer #1: No

Reviewer #2: **Yes: **Marina V Viana

---

## [Author Response · Author response to Decision Letter 1]

2 Sep 2022

September 2 2022

Dr. Rishabh Charan Choudhary 

PLOS ONE

Manuscript ID: PONE-D-21-39517

Title: Impact of early nutrition and feeding route on clinical outcomes of neurocritically ill patients

Dear Dr. Rishabh Charan Choudhary

Thank you very much for your letter and for the helpful comment from the reviewer. We appreciate the opportunity to resubmit our revised manuscript entitled “Impact of early nutrition and feeding route on clinical outcomes of neurocritically ill patients”. As always, you and your editorial staff have again provided us with a comprehensive and prompt review. Many of the valuable and constructive points that the reviewers pointed out were well taken by all the authors. After going over the reviewer’s comments, my colleagues and I have performed additional investigation and made some revisions in hopes of improving our paper. The revised and added portions of the manuscript are stated in the “Response to Reviewers” and are underlined and highlighted in the revised manuscript for your convenience.

We hope that, upon review, our study will be found to be meritorious of publication in the PLOS ONE.

Yours sincerely,

Jeong-Am Ryu, M.D., Ph.D.

Department of Critical Care Medicine and Department of Neurosurgery, Samsung Medical Center, Sungkyunkwan University School of Medicine, 81 Irwon-ro, Gangnam-gu, Seoul 06351, Republic of Korea 

Tel: 82-2-3410-6399, Fax: 82-2-2148-7088

E-mail: lamyud.ryu@samsung.com

Response to Reviewers

Reviewer #1: 

Reviewer #1: Second Review

Thank you for the revised manuscript. It is much improved.

Concerns:

The legend for Table 2 clarifies the comparison. However, since there are three groups being compared it would seem that comparison of continuous variables would have required ANOVA rather than Student’s t-test. If this was done, please add this to the description of the data analysis. If this was not done, please explain what method of analysis was used to compare data for continuous variables in three groups.

In Figure 2, there are lines for the IPTW analysis, but you reported removing this analysis from the manuscript. It should be removed from the Figure as well.

Finally, I previously recommended that you compare the quantity and the composition of the nutrition provided but you indicated that these data are not available. You also indicated that you mentioned this as a limitation in your discussion section. I would encourage you to expand upon this in the discussion. Because the data for the quantity and the composition of the nutrition provided were not available, you are only comparing the timing of initiating nutrition by either the enteral or the parenteral route. However, the quantity of nutrition may have an impact on the findings of your study. Some previous studies have shown a difference in outcomes based on calories delivered, based on protein delivered, and some on both calories and protein delivered. This is an unfortunate deficit in your study.

R1. We apologize for the lack of clear statement. We compared late nutrition, EPN and EEN in original manuscript. In table 2, we used one-way analysis of variance (ANOVA) for continuous variables. We performed ANOVA analysis using the tableone package and the moonBook package of R program. 

We added the following sentences in the Methods section of the revised manuscript

Line 90-92: Data were compared using Student’s t-test and one-way analysis of variance for continuous variables and Chi-square test or Fisher’s exact test for categorical variables.

R2. In revised figure 2, there was no line for IPTW. We showed the line for IPTW only in the first rebuttal letter for your comprehension. 

R3. We agree with the reviewer’s comment. Unfortunately, the amount of EEN or EPN calorie intake for patients in the early stage was not considered in this study due to its retrospective nature. As your recommendation, we added these limitations in the discussion section of the revised manuscript. 

Line 244-253: Adequate calorie and protein intake is important for recovery in critically ill patients [33,34]. Adequate nutritional support may also be important in neurocritically ill patients. The patients with severe traumatic brain injury have increased energy expenditure usually increase by 87%–200% above the usual requirement and may be elevated for 30 days due to metabolic changes [26,35,36]. In addition, systemic catabolic change could lead to hyperglycemia, protein wasting, and increased calorie demands [26,35]. Therefore, optimized calorie and protein supply is also important for acute brain injury patients. However, energy expenditure, calorie and protein intake were not considered in this study. Accurate analysis of calorie and protein supply and energy expenditure may be necessary to investigate the relationship between early nutrition and clinical outcomes in neurocritically ill patients.

Ref 26. Ohbe H, Jo T, Matsui H, Fushimi K, Yasunaga H. Early enteral nutrition in patients with severe traumatic brain injury: a propensity score-matched analysis using a nationwide inpatient database in Japan. Am J Clin Nutr. 2020; 111: 378-84. https://doi.org/10.1093/ajcn/nqz290 PMID: 31751450

Ref 33. Yamamoto S, Allen K, Jones KR, Cohen SS, Reyes K, Huhmann MB. Meeting Calorie and Protein Needs in the Critical Care Unit: A Prospective Observational Pilot Study. Nutr Metab Insights. 2020; 13: 1178638820905992. https://doi.org/10.1177/1178638820905992 PMID: 32153344

Ref 34. Hartl WH, Kopper P, Bender A, Scheipl F, Day AG, Elke G, et al. Protein intake and outcome of critically ill patients: analysis of a large international database using piece-wise exponential additive mixed models. Crit Care. 2022; 26: 7. https://doi.org/10.1186/s13054-021-03870-5 PMID: 35012618

Ref 35. Abdullah MI, Ahmad A, Syed Saadun Tarek Wafa SWW, Abdul Latif AZ, Mohd Yusoff NA, Jasmiad MK, et al. Determination of calorie and protein intake among acute and sub-acute traumatic brain injury patients. Chin J Traumatol. 2020; 23: 290-4. https://doi.org/10.1016/j.cjtee.2020.04.004 PMID: 32423779

Ref 36. Kurtz P, Rocha EEM. Nutrition Therapy, Glucose Control, and Brain Metabolism in Traumatic Brain Injury: A Multimodal Monitoring Approach. Front Neurosci. 2020; 14: 190. https://doi.org/10.3389/fnins.2020.00190 PMID: 32265626

We thank the reviewer for valuable comments. Addressing them fully has significantly strengthened the manuscript. 

Reviewer #2: 

Reviewer #2: Dear authors,

Thank you for your answers.

I do believe the manuscript is improved and all questions have been adequately answered.

R. We thank the reviewer for valuable comments. Addressing them fully has significantly strengthened the manuscript.

---

## [Editor Report · Decision Letter 2]

29 Sep 2022

PONE-D-21-39517R2Impact of early nutrition and feeding route on clinical outcomes of neurocritically ill patientsPLOS ONE

Dear Dr. Ryu,

Thank you for submitting your revised version to PLOS ONE. I am pleased to let you know that your paper has been accepted for the publication however it still needs minor corrections before final acceptance. Please check the response belowReviewer's comment:

The legend for Table 2 clarifies the comparison. However, since there are three groups being compared it would seem that comparison of continuous variables would have required ANOVA rather than Student’s t-test. If this was done, please add this to the description of the data analysis. If this was not done, please explain what method of analysis was used to compare data for continuous variables in three groups.

In Figure 2, there are lines for the IPTW analysis, but you reported removing this analysis from the manuscript. It should be removed from the Figure as well.

Finally, I previously recommended that you compare the quantity and the composition of the nutrition provided but you indicated that these data are not available. You also indicated that you mentioned this as a limitation in your discussion section. I would encourage you to expand upon this in the discussion. Because the data for the quantity and the composition of the nutrition provided were not available, you are only comparing the timing of initiating nutrition by either the enteral or the parenteral route. However, the quantity of nutrition may have an impact on the findings of your study. Some previous studies have shown a difference in outcomes based on calories delivered, based on protein delivered, and some on both calories and protein delivered. This is an unfortunate deficit in your study.

We look forward to receiving your revised manuscript.

Kind regards,

Rishabh Charan Choudhary

Academic Editor

PLOS ONE
---

## [Author Response · Author response to Decision Letter 2]

25 Oct 2022

September 2 2022

Dr. Rishabh Charan Choudhary 

PLOS ONE

Manuscript ID: PONE-D-21-39517

Title: Impact of early nutrition and feeding route on clinical outcomes of neurocritically ill patients

Dear Dr. Rishabh Charan Choudhary

Thank you very much for your letter and for the helpful comment from the reviewer. We appreciate the opportunity to resubmit our revised manuscript entitled “Impact of early nutrition and feeding route on clinical outcomes of neurocritically ill patients”. As always, you and your editorial staff have again provided us with a comprehensive and prompt review. Many of the valuable and constructive points that the reviewers pointed out were well taken by all the authors. After going over the reviewer’s comments, my colleagues and I have performed additional investigation and made some revisions in hopes of improving our paper. The revised and added portions of the manuscript are stated in the “Response to Reviewers” and are underlined and highlighted in the revised manuscript for your convenience.

We hope that, upon review, our study will be found to be meritorious of publication in the PLOS ONE.

Yours sincerely,

Jeong-Am Ryu, M.D., Ph.D.

Department of Critical Care Medicine and Department of Neurosurgery, Samsung Medical Center, Sungkyunkwan University School of Medicine, 81 Irwon-ro, Gangnam-gu, Seoul 06351, Republic of Korea 

Tel: 82-2-3410-6399, Fax: 82-2-2148-7088

E-mail: lamyud.ryu@samsung.com

Response to Reviewers

Reviewer #1: 

Reviewer #1: Second Review

Thank you for the revised manuscript. It is much improved.

Concerns:

The legend for Table 2 clarifies the comparison. However, since there are three groups being compared it would seem that comparison of continuous variables would have required ANOVA rather than Student’s t-test. If this was done, please add this to the description of the data analysis. If this was not done, please explain what method of analysis was used to compare data for continuous variables in three groups.

In Figure 2, there are lines for the IPTW analysis, but you reported removing this analysis from the manuscript. It should be removed from the Figure as well.

Finally, I previously recommended that you compare the quantity and the composition of the nutrition provided but you indicated that these data are not available. You also indicated that you mentioned this as a limitation in your discussion section. I would encourage you to expand upon this in the discussion. Because the data for the quantity and the composition of the nutrition provided were not available, you are only comparing the timing of initiating nutrition by either the enteral or the parenteral route. However, the quantity of nutrition may have an impact on the findings of your study. Some previous studies have shown a difference in outcomes based on calories delivered, based on protein delivered, and some on both calories and protein delivered. This is an unfortunate deficit in your study.

R1. We apologize for the lack of clear statement. We compared late nutrition, EPN and EEN in original manuscript. In table 2, we used one-way analysis of variance (ANOVA) for continuous variables. We performed ANOVA analysis using the tableone package and the moonBook package of R program. 

We added the following sentences in the Methods section of the revised manuscript

Line 90-92: Data were compared using Student’s t-test and one-way analysis of variance for continuous variables and Chi-square test or Fisher’s exact test for categorical variables.

R2. In revised figure 2, there was no line for IPTW. We showed the line for IPTW only in the first rebuttal letter for your comprehension. 

R3. We agree with the reviewer’s comment. Unfortunately, the amount of EEN or EPN calorie intake for patients in the early stage was not considered in this study due to its retrospective nature. As your recommendation, we added these limitations in the discussion section of the revised manuscript. 

Line 244-253: Adequate calorie and protein intake is important for recovery in critically ill patients [33,34]. Adequate nutritional support may also be important in neurocritically ill patients. The patients with severe traumatic brain injury have increased energy expenditure usually increase by 87%–200% above the usual requirement and may be elevated for 30 days due to metabolic changes [26,35,36]. In addition, systemic catabolic change could lead to hyperglycemia, protein wasting, and increased calorie demands [26,35]. Therefore, optimized calorie and protein supply is also important for acute brain injury patients. However, energy expenditure, calorie and protein intake were not considered in this study. Accurate analysis of calorie and protein supply and energy expenditure may be necessary to investigate the relationship between early nutrition and clinical outcomes in neurocritically ill patients.

Ref 26. Ohbe H, Jo T, Matsui H, Fushimi K, Yasunaga H. Early enteral nutrition in patients with severe traumatic brain injury: a propensity score-matched analysis using a nationwide inpatient database in Japan. Am J Clin Nutr. 2020; 111: 378-84. https://doi.org/10.1093/ajcn/nqz290 PMID: 31751450

Ref 33. Yamamoto S, Allen K, Jones KR, Cohen SS, Reyes K, Huhmann MB. Meeting Calorie and Protein Needs in the Critical Care Unit: A Prospective Observational Pilot Study. Nutr Metab Insights. 2020; 13: 1178638820905992. https://doi.org/10.1177/1178638820905992 PMID: 32153344

Ref 34. Hartl WH, Kopper P, Bender A, Scheipl F, Day AG, Elke G, et al. Protein intake and outcome of critically ill patients: analysis of a large international database using piece-wise exponential additive mixed models. Crit Care. 2022; 26: 7. https://doi.org/10.1186/s13054-021-03870-5 PMID: 35012618

Ref 35. Abdullah MI, Ahmad A, Syed Saadun Tarek Wafa SWW, Abdul Latif AZ, Mohd Yusoff NA, Jasmiad MK, et al. Determination of calorie and protein intake among acute and sub-acute traumatic brain injury patients. Chin J Traumatol. 2020; 23: 290-4. https://doi.org/10.1016/j.cjtee.2020.04.004 PMID: 32423779

Ref 36. Kurtz P, Rocha EEM. Nutrition Therapy, Glucose Control, and Brain Metabolism in Traumatic Brain Injury: A Multimodal Monitoring Approach. Front Neurosci. 2020; 14: 190. https://doi.org/10.3389/fnins.2020.00190 PMID: 32265626

We thank the reviewer for valuable comments. Addressing them fully has significantly strengthened the manuscript. 

Reviewer #2: 

Reviewer #2: Dear authors,

Thank you for your answers.

I do believe the manuscript is improved and all questions have been adequately answered.

R. We thank the reviewer for valuable comments. Addressing them fully has significantly strengthened the manuscript.

---

## [Decision Letter · Decision Letter 3]

14 Mar 2023

Impact of early nutrition and feeding route on clinical outcomes of neurocritically ill patients

PONE-D-21-39517R3

Dear Dr. Ryu,

We’re pleased to inform you that your manuscript has been judged scientifically suitable for publication and will be formally accepted for publication once it meets all outstanding technical requirements.

Kind regards,

Raphael Cinotti, MD, PhD

Academic Editor

PLOS ONE

Additional Editor Comments (optional):

Reviewers' comments:

Reviewer's Responses to Questions

**Comments to the Author**

1. If the authors have adequately addressed your comments raised in a previous round of review and you feel that this manuscript is now acceptable for publication, you may indicate that here to bypass the “Comments to the Author” section, enter your conflict of interest statement in the “Confidential to Editor” section, and submit your "Accept" recommendation.

Reviewer #1: All comments have been addressed

2. Is the manuscript technically sound, and do the data support the conclusions?

Reviewer #1: Yes

3. Has the statistical analysis been performed appropriately and rigorously? 

Reviewer #1: Yes

4. Have the authors made all data underlying the findings in their manuscript fully available?

Reviewer #1: Yes

5. Is the manuscript presented in an intelligible fashion and written in standard English?

Reviewer #1: Yes

6. Review Comments to the Author

Reviewer #1: Thank you for the revised manuscript. It is much improved, and all my questions have been answered satisfactorily.

7. PLOS authors have the option to publish the peer review history of their article (what does this mean?). If published, this will include your full peer review and any attached files.

Reviewer #1: No

---

## [Editor Report · Acceptance letter]

16 Mar 2023

PONE-D-21-39517R3 

Impact of early nutrition and feeding route on clinical outcomes of neurocritically ill patients 

Dear Dr. Ryu:

I'm pleased to inform you that your manuscript has been deemed suitable for publication in PLOS ONE. Congratulations! Your manuscript is now with our production department. 

Kind regards, 

on behalf of

Pr. Raphael Cinotti 

Academic Editor

PLOS ONE